# Exploring the Expression of Pro-Inflammatory and Hypoxia-Related MicroRNA-20a, MicroRNA-30e, and MicroRNA-93 in Periodontitis and Gingival Mesenchymal Stem Cells under Hypoxia

**DOI:** 10.3390/ijms231810310

**Published:** 2022-09-07

**Authors:** Alejandra Chaparro, Mauricio Lozano, Dominique Gaedechens, Carolina López, Daniela Albers, Marcela Hernández, Andrés Pascual, José Nart, Carlos E. Irarrazabal

**Affiliations:** 1Department of Oral Pathology and Conservative Dentistry, Periodontics, Faculty of Dentistry, and Centro de Investigación e Innovación Biomédica, Universidad de Los Andes, Av. Plaza 2501, Las Condes, Santiago 7620157, Chile; 2Integrative and Molecular Physiology Laboratory, Physiology Program, Centro de Investigación e Innovación Biomédica, Universidad de los Andes, Av. Plaza 2501, Las Condes, Santiago 7620157, Chile; 3Department of Statistics, School of Dentistry, Faculty of Science, Universidad Mayor, San Pio X 2422, Providencia, Santiago 7510041, Chile; 4Department of Oral Pathology and Medicine, Faculty of Dentistry, University of Chile, Sergio Livingstone Pohlhammer 943, Independencia, Santiago 8380492, Chile; 5Laboratory of Periodontal Biology, Faculty of Dentistry, University of Chile, Sergio Livingstone Pohlhammer 943, Independencia, Santiago 8380492, Chile; 6Department of Periodontology, School of Dentistry, Universitat Internacional de Catalunya, Josep Trueta s/n 08194, Sant Cugat del Vallés, 08017 Barcelona, Spain; 7Faculty of Medicine, Universidad de los Andes, Av. Plaza 2501, Las Condes, Santiago 7620157, Chile

**Keywords:** periodontitis, hypoxia, miRNAs, HIF-1α, NFAT5

## Abstract

Hypoxia associated with inflammation are common hallmarks observed in several diseases, and it plays a major role in the expression of non-coding RNAs, including microRNAs (miRNAs). In addition, the miRNA target genes for hypoxia-inducible factor-1α (HIF-1α) and nuclear factor of activated T cells-5 (NFAT5) modulate the adaptation to hypoxia. The objective of the present study was to explore hypoxia-related miRNA target genes for HIF-1α and NFAT5, as well as miRNA-20a, miRNA-30e, and miRNA-93 expression in periodontitis versus healthy gingival tissues and gingival mesenchymal stem cells (GMSCs) cultured under hypoxic conditions. Thus, a case-control study was conducted, including healthy and periodontitis subjects. Clinical data and gingival tissue biopsies were collected to analyze the expression of miRNA-20a, miRNA-30e, miRNA-93, HIF-1α, and NFAT5 by qRT-PCR. Subsequently, GMSCs were isolated and cultured under hypoxic conditions (1% O_2_) to explore the expression of the HIF-1α, NFAT5, and miRNAs. The results showed a significant upregulation of miRNA-20a (*p* = 0.028), miRNA-30e (*p* = 0.035), and miRNA-93 (*p* = 0.026) in periodontitis tissues compared to healthy gingival biopsies. NFAT5 mRNA was downregulated in periodontitis tissues (*p* = 0.037), but HIF-1α was not affected (*p* = 0.60). Interestingly, hypoxic GMSCs upregulated the expression of miRNA-20a and HIF-1α, but they downregulated miRNA-93e. In addition, NFAT5 mRNA expression was not affected in hypoxic GMSCs. In conclusion, in periodontitis patients, the expression of miRNA-20a, miRNA-30e, and miRNA-93 increased, but a decreased expression of NFAT5 mRNA was detected. In addition, GMSCs under hypoxic conditions upregulate the HIF-1α and increase miRNA-20a (*p* = 0.049) expression. This study explores the role of inflammatory and hypoxia-related miRNAs and their target genes in periodontitis and GMSCs. It is crucial to determine the potential therapeutic target of these miRNAs and hypoxia during the periodontal immune–inflammatory response, which should be analyzed in greater depth in future studies.

## 1. Introduction

In periodontitis, the subgingival periodontal biofilm exhibits a wide range of mechanisms that can affect the host’s immune activation, involving altered gene expression by epigenetic modifications [1,2]. An example is the well-recognized association between the interleukin-1 genotype and periodontitis severity [3]. In addition, the role of other inflammatory cytokines, including IL-4, IL-6, IL-10, IL-17, and IL-37, their polymorphisms, receptors, and signaling pathways are involved in the pathogenesis of periodontitis and the clinical severity of the disease [4,5,6,7,8]. In this sense, it should be emphasized that the levels of these pro-inflammatory mediators seem to be extremely important, as they are in the long-term maintenance of gingival health and the stability of the result of periodontal or implant therapy [9]. It is, therefore, essential to understand which mechanisms are involved at the basis of this process to control the inflammatory and destructive response. Moreover, the growth and maturation of anaerobic and dysbiotic subgingival biofilm could contribute to lower oxygen levels in periodontitis tissues [10], which are also affected by disrupted microcirculation and increased leukocyte infiltration [11,12]. Thus, oxygen supply and consumption in periodontal tissues may be significantly imbalanced and, in response to hypoxia, which are activated by different transcription factors and several non-coding RNAs as microRNAs (miRNAs).

Hypoxia upregulates the transcription factors, such as hypoxia inducible factor-1α (HIF-1α) and Nuclear Factor of Activated T cells-5 (NFAT5) and is critical in regulating cellular responses to hypoxia [13,14]. Interestingly, HIF-1α is almost undetectable under normoxic conditions, but its levels increase in a hypoxic environment [14,15,16]. Furthermore, it is a crucial transcription factor for osteoclast differentiation [17,18]. Moreover, recent evidence suggests that a hypoxic environment and HIF-1α are involved in periodontal inflammatory responses [19,20,21], angiogenesis in mesenchymal stem cells [22], osteoclast formation [23], and the activation of several microRNAs (miRNAs) implicated in essential cell functions and epigenetic modifications [24]. In addition, we have shown that NFAT5 participates in the hypoxia response, but it is independent of HIF-1α [25] and switches the upregulation of inducible-form nitric oxidases synthase by hypoxia [26]. However, there is no evidence of the expression of NFAT5 in the context of periodontal inflammation.

Small and non-coding RNAs, including miRNAs can modify the gene expression of several genes at the same time [27,28,29,30]. Most miRNAs bind imperfectly to their target sequence and selectively repress the translational process, often promoting gene silencing [27,28], controlling the expression of several genes [29]. Among them, miR-20a is downregulated by hypoxia in nasopharyngeal carcinoma, and the miR-17-92 cluster (miRNA-20a) can suppress HIF-1α, suggesting that miRNA-20a is a crucial factor in the hypoxia response [30]. Thus, miRNAs could be responsible for orchestrating several biologic processes, including antibacterial [31,32], anti-inflammatory, and antioxidative mechanisms, as well as being epigenetic regulators [29,33]. MiRNA-93 is also upregulated by hypoxia [34,35]. Indeed, miRNA-93 has the same “seed region” as miRNA-20a (GCACUUU), suggesting that its up-regulation in periodontitis might inhibit the same hypoxia-response genes [35]. 

The expression of miRNAs in periodontal disease has been studied previously [36,37,38,39,40,41,42,43,44]. In this context, miRNAs might mediate endotoxin tolerance through the modulation of the mitogen-activated protein kinase (MAPK) signaling pathway, increasing the sensibility of Toll-like receptors (TLRs) to lipopolysaccharide (LPS), or targeting the nuclear factor-kappa B (NF-kB) signaling pathway in response to bacterial stimuli [31,32,36]. Further, miRNAs are critical regulators in bone homeostasis, affecting osteoclast activity, osteogenic lineage, and stem cells through positive feedback loops [39,40,41,42,43]. However, although various studies have identified miRNAs in periodontal tissues, their essential role in periodontitis pathogenesis needs to be clarified [42,44,45,46,47,48,49].

Intriguingly, miRNA-146a exhibits a crucial function in the negative regulation of the immune response, its dysregulation has been associated with several inflammatory diseases, and it seems to play a functional, key role in periodontitis [32,36,46,48]. Current studies suggest the upregulation of some miRNAs in periodontitis tissues compared to the healthy periodontium [37,38,41,42,43,44,45]. As an example, the overexpression of miRNA-146a in periodontitis patients was also accompanied by a decrease in the release of pro-inflammatory cytokines (TNF-a, IL-1B, and IL-6), suggesting that the elevation of miRNA-146a regulates pro-inflammatory cytokines through a negative feedback loop [31,32,36]. In addition, other miRNAs, including miRNA-30e, miRNA-130a, miRNA-142-3p, miRNA-210, and miRNA-223 (which is involved in osteoclastogenesis) are significantly increased in periodontitis [37,38,41,42,43,44,47]. 

Understanding the functional roles of hypoxia and miRNAs in the pathogenesis of periodontitis is relevant due to their strong potential as therapeutic targets in inflammatory disease treatment and bone regeneration. Most likely, miRNA therapeutics hold great promise for the future of periodontal therapy based on their ability to modulate the immune response against infections [41,42,44,47]. The present study aimed to explore the expression of inflammatory hypoxia-related miRNAs (miRNA-20a, miRNA-30e, and miRNA-93), and miRNA target genes, HIF-1α and NFAT5, in periodontitis and healthy gingival tissues. Additionally, we aimed to analyze the effect of hypoxia on these miRNAs, HIF-1α, and NFAT5 expression in gingival mesenchymal stem cells (GMSCs). 

## 2. Results

### 2.1. MiRNA Expression in Healthy and Periodontal Disease Biopsies

According to previously published data, the selected, studied miRNAs are involved in hypoxia and inflammatory mechanisms. We selected these miRNAs based on their targeted sequence homology with the 3′UTR region of NFAT5 and HIF-1α, using a bioinformatic approach (miRbase and TargetScan). The expressions of miRNA-20a, miRNA-30e, miRNA-93, HIF-1α, and NFAT5 were assessed by qRT-PCR and normalized with U6 small nucleolar RNA abundance for miRNA and 18S for mRNA. In the periodontitis biopsies, we observed a significant increase of miRNA-20a (12.8 times, *p* = 0.028), miRNA-30e (2.1 times, *p* = 0.035), and miRNA-93 (4 times, *p* = 0.026) expression compared with healthy gingival tissues (Figure 1). An association between miRNA-20a (OR = 1.02, CI 95%: 1.002–1.045), miRNA-30e (OR = 1.047, CI 95%: 1.003–1.093), and miRNA-93 (OR = 1.021, CI 95%: 1.002–1.041) with the diagnosis of periodontitis was assessed. Thus, to further understand the diagnostic performance of these miRNAs in periodontitis, we studied the area under ROC curve analysis, which corresponded to 0.84 (with a sensitivity of 80%, and a specificity of 88.89%) for miRNA-20a, 0.86 for miRNA-30e (with a sensitivity of 90%, and a specificity of 66.67%), and the AUC-ROC curve of 0.88 for miRNA-93 (with a sensitivity of 90%, and a specificity of 66.67%) (Figure 2). Higher sensitivity and specificity results for periodontitis diagnoses were obtained for miRNA-93 and miRNA-20a, respectively.

In addition, we analyzed the mRNA expressions of HIF-1α and NFAT5. The concentration of mRNA obtained from NFAT5 was significantly lower in periodontitis biopsies compared to healthy gingival tissue (*p* = 0.037). However, the levels of HIF-1α mRNA were similar in both groups, suggesting that the studied miRNAs (miRNA-20a, miRNA-30e, and miRNA-93) could be involved in the downregulation of NFAT5 in periodontitis tissues.

### 2.2. MiRNA Expression in Gingival Mesenchymal Stem Cells (GMSCs) under Hypoxic Conditions

The GMSCs were positive for CD73, CD90, and CD105; negative for CD11b, CD34, CD45, and HLA-DR; negative for plastic adherence; and positive for differentiation-staining of chondrogenic, osteogenic, and adipogenicity lineages (Figure 3). Under optic microscopy, GMSCs in normal oxygen conditions showed a fibroblast-like, spindle shape and a plastic-adherent property. Under a hypoxic environment, GMSCs maintained their fibroblast-like, spindle shape and plastic-adherent property (baseline, 3, 6, and 12 h) (Figure 3). Subsequently, the effect of hypoxia on the expression of miRNA-20a, miRNA-30e, miRNA-93, HIF-1α, and NFAT5 in GMSCs was assessed. The results showed that the hypoxic condition downregulated miRNA-30e at 3 h of hypoxia (*p* = 0.015) and upregulated miRNA-20a at 6 h of hypoxic (*p* = 0.049) condition in GMSCs (Figure 4).

### 2.3. Effect of Hypoxia in the Expression of HIF-1α and NFAT5 in GMSC

The expression of HIF-1α and NFAT5 was analyzed in GMSCs culture cells. In standard oxygen conditions (21% O_2_), HIF-1α expression was scarce. However, the HIF-1α expression was significantly upregulated at 3 h of hypoxia (1% O_2_), which decreased towards 12 h, suggesting that the miRNA-30a downregulation at 3 h could be involved in the HIF-1α upregulation at the same time (Figure 5). In addition, the miRNA-20a upregulation at 6 h could be associated with the HIF-1α downregulation. Furthermore, the NFAT5 expression was not modified by the hypoxic condition (Figure 5), suggesting that these miRNAs are not involved in controlling NFAT5 expression under hypoxia in GMSCs.

## 3. Discussion

The present study shows an upregulated expression of miRNA-20a, miRNA-30e, and miRNA-93 in gingival tissues from periodontitis patients compared with the gingival biopsies from healthy controls. Furthermore, our results suggest that NFAT5, but not HIF-1α mRNA, was downregulated in periodontitis patients, suggesting that the activation of miRNAs is potentially associated with NFAT5 expression in periodontal tissues. Otherwise, the GMSCs cultured under hypoxic conditions significantly decreased the levels of miRNA-30e and increased the HIF-1α expression, suggesting that miRNA-30a could participate in HIF-1α upregulation during hypoxia. These results are noteworthy because the hypoxic condition could increase alveolar bone resorption in periodontitis, and probably, this low-oxygen tension in periodontal tissues contributes to the loss of periodontal support tissues [50]. Moreover, hypoxia could be one of the main contributors to the miRNA’s master regulation in inflamed periodontal tissues observed in periodontitis [24,30,44,47,51]. Thus, the main findings of this exploratory study are: 1. A significant increase in the expression of miRNA-20a, miRNA-30e, and miRNA-93 in periodontitis; 2. A decreased expression of the NFAT5 mRNA, and a slight decrease in HIF-1α mRNA. Thus, the upregulation of these inflammatory and hypoxia-related miRNAs, accompanied by an NFAT5 mRNA downregulation, could be involved in periodontitis pathogenesis, and this should be examined in greater depth in further studies. 

In this sense, miRNAs have been proposed as gene-expression-regulator molecules and potential biomarkers of periodontitis [41,42,44,47,49]. The serum levels of miRNAs in periodontitis patients have also been characterized, suggesting the upregulation of miRNA-664a, miRNA-501, and miRNA-21-3p, among others [52]. Similarly, miRNA-146a and miRNA-155 in gingival crevicular fluid and miRNA-21-3p in periodontal tissues are associated with periodontitis severity [48,53,54]. Other miRNAs, such as miRNA-130a, miRNA-301a, miRNA-520d, and miRNA-548a, were upregulated in inflamed gingival tissues and periodontitis [52,53]. Our findings support the existence of an increased expression of miRNA-20a, miRNA-30e, and miRNA-93 in the periodontal tissues of periodontitis subjects.

A member of the miR-17-92 cluster, miRNA-20a is one of the most extensively studied families. The members of this family play essential roles in tissue and organ development, and they are closely associated with tumors, autoimmune diseases, and osteogenesis [28,29,41,42]. In addition, the overexpression of miRNA-20a has been observed in inflamed gingival tissues [53,54]. In the case of the miRNA-30e, it has been reported to be associated with cancer [55], cardiac dysfunction [56], and elevations of the innate immune response during viral infections and in autoimmune diseases [57]. In the periodontal disease context, a decreased expression of miRNA-30e in inflamed tissues compared with healthy gingiva was reported [54]. Furthermore, miRNA-93e is expressed in normal and pathological scenarios and has been involved in osteogenesis [58,59]. It seems that the increase of miRNA-93 could inhibit osteogenic differentiation by targeting bone morphogenetic protein-2 [59]. Here, the present results suggest the upregulation of miRNA-20a, miRNA-30e, and miRNA-93 in periodontitis cases.

Additionally, miRNAs are also affected by a hypoxic environment [24,30,34,35,51] and may be regulated in an HIF-1-dependent or -independent manner by repressing the expression of HIF-1α, HIF-1β, or a plethora of possible downstream targets to affect hypoxic responses [13,14,19,20]. Alternatively, miRNAs may initiate new gene-expression programs to enable adaptation to long-term hypoxic stress [24,30,34,35,51]. Understanding the regulatory mechanisms of miRNAs related to hypoxic responses is crucial to recognizing the pathophysiology of various chronic inflammatory diseases [51], including periodontitis [52,53,54].

Interestingly, during O_2_ deprivation, many cellular responses are primarily regulated by hypoxia-inducible factors (HIFs), as well as in pathological settings which involve the inflammation process [60]. HIF-1α is a critical transcription-activator related to oxygen homeostasis [14,60]. In gingival biopsies from periodontitis subjects (in a hypoxic pathological context), the fibroblast-like cells and infiltrating inflammatory cells overexpress HIF-1α, suggesting that the HIF pathway is involved in the inflammatory response of human periodontal tissue [10,11,12,61]. In addition, high HIF-1α and NFATc1 levels were detected in severe periodontitis and gingival crevicular fluid samples of severe periodontitis patients [20], suggesting a role for the TNF-α/HIF-1α/VEGF pathway in the pathogenesis of periodontitis [61,62]. Rather, NFAT5 plays a central role in inducible gene expression during the immune response stimulating the expression of various pro-inflammatory cytokines [63]. Furthermore, the stimulus of *Porphyromonas gingivalis* on human monocyte cells via Toll-like receptors (TLRs) increased miRNA-132 and inhibited NFAT5 expression [64]. However, in the present study, we found a downregulation of NFAT5 expression in periodontitis biopsies. In addition, in hypoxic GMSCs, we found a downregulation of miRNA-93e without changes in NFAT5 expression. Together, both observations suggest that NFAT5 is downregulated in periodontitis, independently of the hypoxic condition, which could be associated with the upregulated expression of miRNA-20a, miRNA-30e, and miRNA-93 as a mechanism involved in the regulation of the inflammatory response and osteogenic inhibition. In fact, a recent study has suggested that the downregulation of NFAT5 is related to a significantly inhibited osteogenic differentiation in cementoblasts, which results in the activation of various signaling pathways, including those of Erkl/2, JNK, p38, PI3K-Akt, NF-ĸB, and Wnt/β-catenin [65].

We are optimistic that advances in miRNAs research and other non-coding RNAs could improve our understanding of the mechanisms underlying the pathogenesis of chronic inflammatory diseases affected by hypoxia, such as periodontitis. In addition, miRNA research opens new possibilities for new, personalized diagnostic tools based on identifying miRNA profiles in tissues or oral fluids such as gingival crevicular fluid or saliva. In the current study, we found an increased expression of miRNA-20a, miRNA-30e, miRNA-93, and decreased levels of NFAT5 mRNA in periodontitis biopsies, which should be investigated in greater depth. The levels of these inflammatory hypoxia-related miRNAs could probably be used as biomarkers of the inflammatory status and activity of periodontitis in the future. In addition, the upregulation of HIF-1α and miRNA-20a in GMSCs under hypoxia also could contribute to clarifying the role of hypoxia and miRNAs in inflamed periodontal tissues. The present results represent an initial approach to describing some hypoxia- and inflammatory-related miRNAs in the context of periodontitis and assessing the impact of these miRNAs on the pathogenesis and epigenetic modifications that can occur in periodontitis. As a weakness of the present study, we must mention the limited sample size. Thus, our results should be interpreted cautiously and validated in a more extensive study.

## 4. Materials and Methods

### 4.1. Study Design

An exploratory case-control study was conducted. The population in the study consisted of 19 subjects (10 patients with periodontitis and 9 gingivally healthy control subjects), randomly selected from the Universidad de Los Andes Health Care Centre, Santiago, Chile. Both groups were paired for age, gender, body mass index, and socioeconomic level. Clinical and anthropometric data were recorded, and a dental evaluation with a full-mouth periodontal exam was performed by a professional qualified in periodontics. All clinically relevant data for the study were stored in a computer database. Patients were excluded if they had a diagnosis of diabetes or cardiovascular disease, smoked, had fewer than 18 teeth, had used systemic or topical antimicrobial/anti-inflammatory therapy during the previous 3 months, or had a history of previous periodontal treatment during the preceding year. All the subjects signed an informed consent which was approved by the Universidad de Los Andes Ethics Committee. 

### 4.2. Sampling Collection

Gingival tissue samples were collected from healthy periodontal subjects via the distal wedge surgical technique with a gingival index (GI) of <1, a periodontal probing depth (PPD) of <3 mm, a clinical attachment loss (CAL) of <1 mm, and a lack of radiographic evidence of alveolar bone loss; fewer than 10% of the sites showed BOP (in the healthy subjects). The periodontitis group was composed of patients with Stage III or IV periodontitis, including a GI of >1, at least 5 sites with PPD ≥ 5 mm, a CAL of ≥3 mm [66], evidence of radiographic bone loss, and an indication of respective periodontal surgery. Under local anesthesia, a sample of approximately 2 × 2 × 1 mm of keratinized gingival tissue was taken, including junctional epithelium, gingival margin, and connective tissue. The gingival tissue biopsies (approximately 12 g) were deepithelialized with a scalpel, leaving only the connective tissue; placed in a sterile tube with Eagle’s medium with 10% qualified fetal bovine serum (FBS) and 1% penicillin, streptomycin, and amphotericin; and transported to the laboratory. 

According to the protocol previously published by our research group, the explants were placed on tissue culture dishes containing a complete medium (α-MEM). Briefly, the explants were minced and maintained in an incubator at 37 °C with humidified air (5% CO_2_). Between days 14 and 21, the cell cultures reached 80% confluence, at which point they were treated with trypsin 1× and subcultured [67,68]. The purity of the isolated gingival mesenchymal stem cells (GMSCs) was determined by the minimal criteria proposed by the International Society for Cellular Therapy: positive for CD73, CD90, and CD105 markers; negative or weakly positive for hematopoietic markers CD11b, CD34, CD45, and HLA-DR; characterized by plastic adherence; and exhibiting positive differentiation staining of chondrogenic, osteogenic, and adipogenicity lineages (Figure 3). For osteogenic differentiation, cells were cultured in α-MEM medium at a density of 50,000 cells-per-well on 24-well cultures plates in media consisting of α-MEM; 10% FBS; 1% penicillin, streptomycin, and dexamethasone (0.1 mM); β-glycerophosphate (10 mM); and ascorbic acid (50 µg/mL). The osteogenic potential differentiation was stained with alizarin red, which distinguishes the presence of calcified deposits in the culture. For adipogenic differentiation, cells were cultured at a density of 30,000 cells-per-well on 24-well culture plates. Upon reaching confluence, and under a differentiation media, the presence of cell drops of lipids was assessed through stains with red oil. Finally, for chondrogenic differentiation, cells were cultured at density of 30,000 cells-per-well on a 24-well cultured plates. After de-induction with the differentiation media, chondrogenic differentiation was evaluated by staining glycosaminoglycans in the culture via the application of Safranin O [67]. For every sample submitted to the differentiation protocol, a control culture was kept in α-MEM. This culture was stained with the same staining test that was used on the experimental cells.

### 4.3. Cell Culture

GMSCs were cultured in a complete medium consisting of α-MEM, 10% fetal bovine serum, and 2% penicillin–streptomycin (Gibco, Life Technologies, New York, NY, USA) in a CO_2_ incubator (Series 8000, Thermo Scientific, Waltham, MA, USA) at 21% O_2_, 37 °C, and 5% CO_2_. Once the GMSCs reached a level of 70–80% of confluence, the cells were incubated at 37 °C in a hypoxic environment (1% O_2_, C-chamber (C-274), Biospheric, New York, NY, USA) for 3, 6, and 12 h, using a balanced Nitrogen-CO_2_ 5%. Normoxic controls were also included. The morphology and confluence of the GMSCs were corroborated using an optical microscope (Motic ae31, Xiamen, China) and photographed (Moticam 2300, 3.0 m pixel USB 2.0, Xiamen, China).

### 4.4. RNA Extraction and qRT-PCR Analysis

RNA isolation and analysis: flash-frozen gingival tissue (25–35 mg) was placed in QIAzol Lysis Reagent and disrupted with a tissue rupture for 30 s. After the dissociation of the nucleoprotein complexes, the homogenized tissue lysates were incubated with chloroform for subsequent phase separation. The homogenate was kept at room temperature and centrifuged. The aqueous phase was placed in ethanol for RNA precipitation. The obtained samples were purified with a RNeasy Mini Kit (Qiagen, Hilden, Germany). Finally, the RNA was eluted with RNase-free water. An extra step was developed: RNA treatment with DNase I free RNase (Promega, Madison, WI, USA) for genomic DNA degradation. RNA was then quantitated using a UV-Vis spectrophotometer (Nanodrop 2000, Thermo Scientific, Waltham, MA, USA). After the isolation and quantitation of the RNA, 1 µg of input RNA was used for a reverse transcription reaction using a miScript II RT Kit (Qiagen). The cDNA obtained from the reverse transcription reactions was stored at −20 °C or immediately used to perform real-time PCR in a Rotor-Gene Q (Qiagen) thermocycler MyScript PCR System (Qiagen). All samples were run in technical duplicate for each reaction as a qRT-PCR analysis. Primers that were gene-specific to hsa-miR-20a, hsa-miR-30e, hsa-miR-93, hsa-miR-128-1, and hsa-miR-186 were purchased (Qiagen). The endogenous expression of the U6 snRNA gene (Qiagen) was used to normalize the samples’ miRNA expressions. Furthermore, a no-template control (NTC) was included in the analysis via the addition of RNase-free water instead of using RNA samples. 

### 4.5. HIF-1α and NFAT5 Expression

GMSCs were harvested with lysis buffer (Complete Lysis-M, ROCHE, Basil, Switzerland) and were then centrifuged (Centrifuge 5415 rpm, Eppendorf, Hamburg, Germany) at 13,000 rpm for 7 min at 4 °C. The supernatant was collected, and protein concentrations were determined using a BCA protein assay (Pierce, Thermo Scientific, Rockford, IL, USA). Western blot analysis was used to analyze the protein expressions using anti-HIF-1α (rabbit polyclonal igg, Cell Signaling, Danvers, MA, USA), anti-NFAT5 (rabbit polyclonal IgG, Thermo Scientific, Waltham, MA, USA), and anti α–Tubulin (mouse monoclonal IgG, ABCAM, San Francisco, CA, USA). The secondary antibodies that we used were Alexa Fluor 750 goat anti-rabbit and Alexa Fluor 680 goat anti-mouse (Molecular Probes, Eugene, OR, USA). Signals were detected with an infrared fluorescent system (Odyssey clx, LI-COR, Lincoln, NE, USA).

### 4.6. Statistics Analysis

All experiments (qRT-PCR and Western blot) with gingival biopsies were independently performed at least 3 times using 3 samples each time. The expressions of the microRNAs, NFAT5, and HIF-1α were analyzed as continuous variables. The descriptive analysis was performed based on the median and interquartile range. The association strength was assessed using a simple logistic regression model. The crude odds ratio (OR) with a 95% confidence interval (CI) was determined. The discrimination performance of the miRNAs for healthy gingival tissues versus that for periodontitis tissues was evaluated after the implementation of a de-regression model through the area under the curve (AUC) of the receiver operating characteristics (ROC) curve. The optimal cut-off points for estimating the Youden index altogether assessed the highest sensitivity and specificity. Data for the miRNA were transformed into each value’s logarithm because these variables had no normal distribution (Shapiro–Wilks test). A *p*-value < 0.05 was considered statistically significant. The statistical analysis was performed using Stata software (version 16.1, Lakeway Drive, College Station, TX, USA).

## Figures and Tables

**Figure 1 ijms-23-10310-f001:**
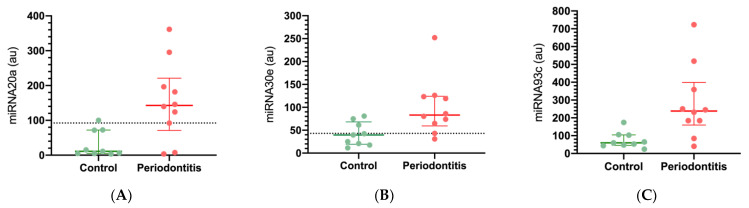
The miRNA-20a, miRNA-30e, and miRNA-93 expressions were upregulated in periodontitis tissues (**A**) miRNA-20a expression (*p* = 0.028); (**B**) miRNA-30e expression (*p* = 0.035); (**C**) miRNA-93c expression (*p* = 0.026). (au): arbitrary units. The relative miRNAs expressions were determined by qRT-PCR and normalized with U6. A suggested threshold of the periodontitis condition is suggested by a dotted lin.

**Figure 2 ijms-23-10310-f002:**
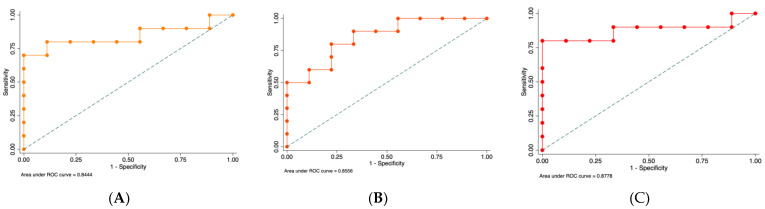
AUC-ROC curve analysis of selected miRNAs: (**A**) miRNA-20a (AUC-ROC: 0.84), (**B**) miRNA-30e (AUC-ROC: 0.85), and (**C**) miRNA-93 (AUC-ROC: 0.87). AUC-ROC: area under the ROC curve.

**Figure 3 ijms-23-10310-f003:**
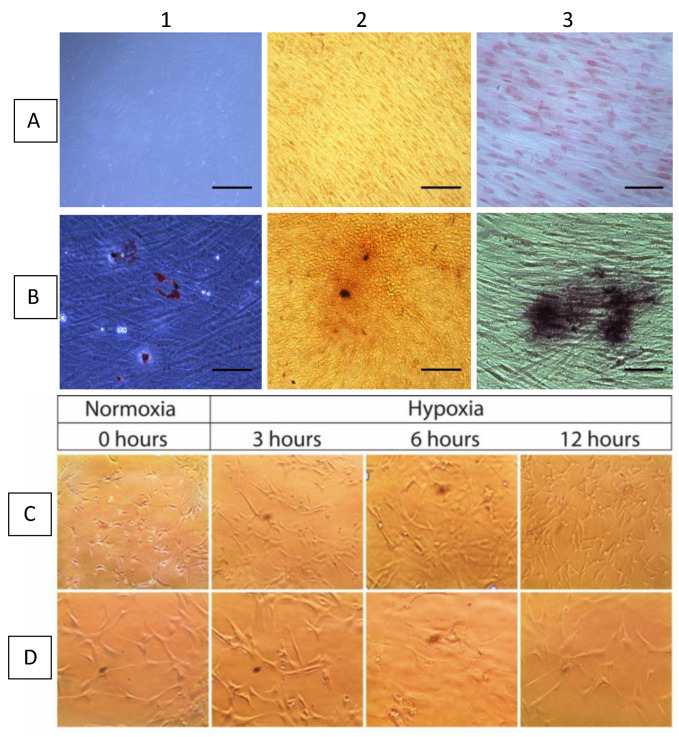
(**A1**–**A3**) Undifferentiated GMSCs control for each lineage; (**B1**) adipogenic, (**B2**) osteogenic, and (**B3**) chondrogenic differentiation of GMSCs; (**C**,**D**) GMSCs under hypoxic environment (baseline, 3, 6 and 12 h) maintained their fibroblast-like, spindle shape and plastic-adherent properties, reaching a slightly higher confluence (**D**) versus normoxic controls (**C**). Scale bar: 50 μm.

**Figure 4 ijms-23-10310-f004:**
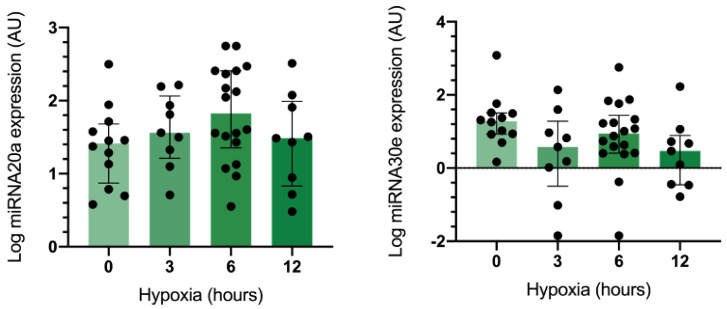
The expression of miRNA-20a was upregulated (*p* = 0.049), and the expression of miRNA-30e was downregulated (*p* = 0.035) in cultures of GMSCs exposed to a hypoxic environment. The relative miRNA expression was determined by qRT-PCR and normalized with U6. (au): arbitrary units.

**Figure 5 ijms-23-10310-f005:**
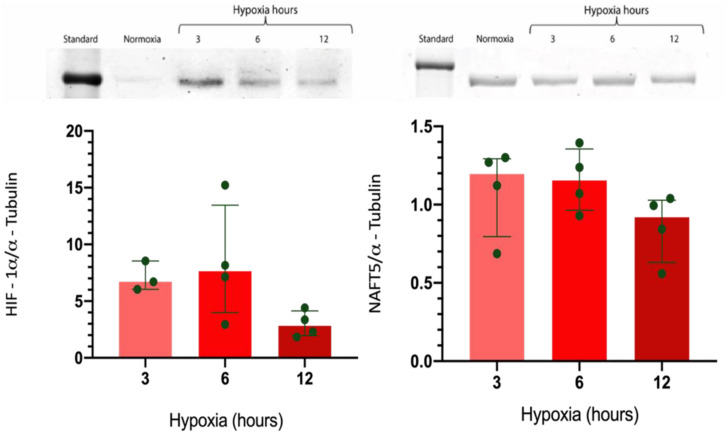
The hypoxic environment (1% oxygen) caused an upregulation of the HIF-1α protein expression and a downregulation of NFAT5 mRNA compared to normoxic conditions (21% oxygen) in GMSC cultures, as analyzed by Western blot (normalized by α-tubulin).

## Data Availability

Not applicable.

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
