# Peer review of "Exploring the Expression of Pro-Inflammatory and Hypoxia-Related MicroRNA-20a, MicroRNA-30e, and MicroRNA-93 in Periodontitis and Gingival Mesenchymal Stem Cells under Hypoxia"

_ijms, 2022, doi:10.3390/ijms231810310_

Round 1
Reviewer 1 Report
The study of Chaparro et al. explores the role of miR-20a, miR-30e and miR-93 in periodontitis and they relations to hypoxic conditions. The expression of selected miRNAs, as well as HIF-1 and NFAT5, was analyzed in periodontal tissue from 10 patients with periodontitis and 9 gingival healthy controls. In addition, the same targets were studied in gingival mesenchymal stem cells (GMSCs) cultured from the same cases under normoxic and hypoxic conditions in order to prove hypothesis on the role of hypoxia in periodontitis-specific changes development.
Major points. The study cohort is too small for reliable association detection and the target genes are selected from published data. The upregulation of hypoxia-related transcripts HIF-1a and NFAT5 was not detected in tissue samples from periodontitis patients, thus further studies on hypoxia-induced changes in GMSCs is of limited meaning. Inclusion of more parameters of hypoxia and wider list of miRNAs might strengthen the study and provide some rationale on the role of hypoxia conditions in periodontitis.
Minor points.
1. It is not clear how many GMSCs were involved in analysis, Methods do not include this information. Also, it is unclear how many replicates were used for qRT-PCR and Western blot experiments. Please provide representative Fig for GMSCs showing trilineage differentiation potential.
2. Row 126-127 - unclear how the numbers for „higher sensitivity and specificity“ were generated.
3. It is unclear why HIF-1a and NFAT5 expression in GMSCs was studied by western blot, while in clinical samples qRT-PCR was used. Application of different methods might explain the discrepancy in data.
4. Upregulation of miR-20a in GMSCs was of borderline significance – 0.049, while the effect of this miRNA on regulation of HIF-1a and NFAT5 is discussed in the Results 2.3 section. Please avoid unsupported speculations in the text.
5. In Fig 1 and Table 1, as well as in Fig 2 and Table 3, the same results are repeated and also they are mentioned in the text. Please avoid repeated presentation of the same data.
6. Please provide more solid explanation on possible mechanisms of NFAT5 down-regulation in periodontitis tissues, instead of stating that “more complex signaling pathways” might exist.
7. Typing errors are left in the text and the text might benefit from language editing.
Table 1 “*The data shown are shown”…
Line 86 “by target the” – by targeting.
Line 321 “1ug” - 1 µg.
...
Author Response
Dear Reviewer.
International Journal of Molecular Sciences
We greatly appreciate your expert editorial guidance and acknowledge the referees for providing a very insightful and constructive review of our manuscript. We feel that we have satisfactorily addressed all the reviewers’ observations and hope that you will now find our manuscript acceptable for publication in the International Journal of Molecular Sciences.
Following the editor and reviewers' comments, our actions are detailed below, and all changes have been highlighted in yellow and bold in the corrected manuscript. The answers to the reviewer’s observations are detailed below.
Dear Reviewer 1:
Major points. The study cohort is too small for reliable association detection, and the target genes are selected from published data. The upregulation of hypoxia-related transcripts HIF-1a and NFAT5 was not detected in tissue samples from periodontitis patients. Thus, further studies on hypoxia-induced changes in GMSCs are of limited meaning. The inclusion of more parameters of hypoxia and a wider list of miRNAs might strengthen the study and provide some rationale for the role of hypoxia conditions in periodontitis.
Answer:
We appreciate all the relevant reviewer`s suggestions that will undoubtedly improve the quality of our article. We have elaborated a detailed response to each comment and criticism of the reviewer. Thus, we have incorporated the changes in the new version of the manuscript.
1.- We agree with the reviewer, and the study cohort is small for a definitive association between miRNA, target genes, and periodontitis. But, with this small group of patients, we found significant differences in the miRNAs expression of miRNA20a, miRNA30e, and miRNA93 in periodontitis patients compared to healthy gingival biopsies. So, we agree with you that these group of miRNAs must be validated in the future in a larger study, and we add this point in the weakness of the study in the discussion with the following paragraph: “As a weakness of the present study, we must mention the limited sample size. Thus, our results should be interpreted with caution and validated in a larger study. The present results represent an initial approach to describing some hypoxic and inflammatory-related miRNAs in periodontitis and GMSCs cultures that encourage analysis of more in-depth miRNAs and hypoxia in the periodontitis context and to assess the impact of these miRNAs in the pathogenesis and epigenetic modifications that could occur in periodontitis in more extensive studies”. In addition, if you consider it necessary, we can delete the results obtained from de logistic regression model and AUC-ROC curve analysis. Furthermore, we observed a downregulation of the NFAT5 mRNA in gingival tissue biopsies (compared with healthy tissue, p-value = 0.037); however, HIF-1α mRNA was similar in both groups. Thus, our hypothesis suggests probably the overexpression of miRNA-20a, miRNA-30e, and miRNA-93 could be involved in the downregulation of NFAT5 and modulated by periodontal inflammation. We totally agree with the reviewer that we need more studies to verify the mRNA-NFAT5 association and to validate the miRNA-gingival inflammation relationship in a larger study, but this is an interesting first step that will be validated in a new study with a wider list of miRNAs and a big sample size. We are very enthusiastic and interested in the research of transcriptomic biomarkers and non-coding RNAs in periodontitis and periimplantitis disease.
Therefore, the contribution of this study is:
- Periodontitis increased the expression of miRNA-20a, miRNA-30e, and miRNA-93
- Periodontitis decreased the NFAT5 mRNA slightly to HIF-1α mRNA.
- Hypoxia in GMSCs cultures does not affect the modulation of miRNA-30e, and miRNA-93, NFAT5, or HIF-1α observed in periodontitis. Only miRNA-20a was increased in periodontitis and in the borderline significance (p=0.049) in hypoxic GMSCs.
- Thus, the miRNAs (miRNA-20a, miRNA-30e, and miRNA-93) upregulation and NFAT5 mRNA downregulation could be interesting mediators involved in the pathogenesis of periodontitis and osteoclastogenesis, but we need validation of these RNAs as well as others for clarifying its role in periodontitis.
Minor points.
1. It is not clear how many GMSCs were involved in the analysis; the Methods do not include this information. Also, it is unclear how many replicates were used for qRT-PCR and Western blot experiments. Please provide representative Fig for GMSCs showing trilineage differentiation potential.
Answer 1- We have added a new paragraph in the Methods section of the new version of the manuscript to include the information on the experimental methodology: “For osteogenic differentiation, cells were cultured in a-MEM medium at a density of 50,000 cells per well on 24-well cultures plates in media consisting of a-MEM, 10% FBS, 1% penicillin, streptomycin, and dexamethasone (0.1mM), b-glycerophophate (10mM) and ascorbic acid (50ug/ml). The osteogenic potential differentiation was stained with alizarin red, which distinguishes the presence of calcified deposits in the culture. For adipogenic differentiation, cells were cultured at a density of 30,000 cells per well on 24-well culture plates. On reaching confluence and under a differentiation media, the presence of cell drops of lipids was assessed through stains with red oil. Finally, for chondrogenic differentiation, cells were cultured at a density of 30,000 cells per well on a 24-well cultured plate. After de induction with the differentiation media, chondrogenic differentiation was evaluated by stained glycosaminoglycans in the culture by the application of safranin O [69]. For every sample submitted to the differentiation protocol, a control culture was kept in a-MEM. This culture was stained with the same staining test as the experimental cells”.
Also, we include the figure of GMSCs showing the trilineage differentiation potential (Figure 3), and in the section on the statistical analysis of the methodology, we include the next paragraph: “All experiments (qRT-PCR and Western blot) with gingival biopsies and cultured GMSCs under a normoxic (21% 02) or hypoxic environment (1% 02), were independently performed at least thrice with 3 samples each”.
2. Row 126-127 - unclear how the numbers for „higher sensitivity and specificity“ were generated.
Answer 2. We have added in the section of the statistical analysis of the methodology, the next paragraph: “Discrimination performance of the miRNAs for healthy gingival tissues versus periodontitis tissues was evaluated after de regression model through the area under the curve (AUC) of the receiver operating characteristics (ROC) curve.The optimal cut-off points to estimate Youden´s Index altogether assessed the highest sensitivity and specificity” in the new version of the manuscript.
3. It is unclear why HIF-1a and NFAT5 expression in GMSCs was studied by western blot, while in clinical samples, qRT-PCR was used. The application of different methods might explain the discrepancy in data.
Answer 3. Yes, the review is right. We did not run Western blot for gingival biopsies (periodontitis and healthy) because the amount of the sample was enough to run miRNA and mRNA experiments, but it was not adequate for Western blot. In the case of GMSCs, we have enough samples to run the Western blot.
4. Upregulation of miR-20a in GMSCs was of borderline significance – 0.049, while the effect of this miRNA on the regulation of HIF-1a and NFAT5 is discussed in the Results 2.3 section. Please avoid unsupported speculations in the text.
Answer 4. We used a more specific description of the miRNA20a, and we corrected it in the new version of the manuscript.
5. In Fig 1 and Table 1, as well as in Fig 2 and Table 3, the same results are repeated and are also mentioned in the text. Please avoid the repeated presentations of the same data.
Answer 5. We eliminate the repeated results of Fig 1 and Table 1, as well as in Fig 2 and 3 in the results section of the new manuscript version.
6. Please provide a more solid explanation of possible mechanisms of NFAT5 down-regulation in periodontitis tissues instead of stating that “more complex signaling pathways” might exist.
Answer 6. We have added a new paragraph in the Discussion section of the new version of the manuscript: “However, in the present study, we found a downregulation of NFTA5 expression in periodontitis biopsies. In hypoxic GMSCs, we also found a downregulation of miRNA-93e without changes in NFAT5 expression. Together, both observations suggest that NFAT5 is downregulated in periodontitis, independently of the hypoxia condition, which could be associated with the upregulated expression of miRNA-20a, miRNA-30e, and miRNA-93, as a mechanism involved in the regulation of the inflammatory response and osteogenic inhibition. In fact, a recent study suggests that the downregulation of NFAT5 is related to a significantly inhibited osteogenic differentiation in cementoblasts, which results in the various signaling pathways activation, including the Erkl/2, JNK, p38, PI3K-Akt, NF-kB, and Wnt/b-catenin”.
7.- Typing errors are left in the text, and the text might benefit from language editing.
Answer 7. We have incorporated the suggestion of the reviewer, and we correct the typing errors and language editing
8.- Table 1 “*The data shown are shown”…
Answer 8. Table 1 was eliminated in the new version of the manuscript.
9.- Line 86 “by target the” – by targeting.
Answer 9. Line 86 was corrected in the new version of the manuscript.
10. Line 321 “1ug” - 1 µg.
Answer 10. Line 321 was corrected in the new version of the manuscript
With Kind Regards
Alejandra Chaparro

Reviewer 2 Report
Dear Authors,
you made a really great work!
However, some improvements are mandatory before acceptance.

Author Response
Dear Reviewer.
International Journal of Molecular Sciences
We greatly appreciate your expert editorial guidance and acknowledge the referees for providing a very insightful and constructive review of our manuscript. We feel that we have been satisfactorily addressed all the reviewers’ observations and hope that you will now find our manuscript acceptable for publication in the International Journal of Molecular Sciences.
Our specific actions are detailed below following the editor and reviewers' comments, and all changes have been highlighted in yellow and bold in the corrected manuscript. The answers to the reviewer’s observations are detailed below
1.-“Also, the role of other inflammatory mediators (IL-4, IL-6, IL-10, IL-17, and IL-37), their polymorphisms, receptors, and signaling pathways are involved in the pathogenesis of periodontitis and the clinical severity of the disease [4 - 8].” it should be emphasized that the levels of these mediators seem to be extremely important, as well as in the long-term maintenance of soft tissue health, also in the stability of the result of implant or periodontal therapy. As emerged from the analysis of the sampling of sulcular and crevicular fluids, the level of inflammation of the peri-implant is always greater, even in a healthy condition, than that of the periodontally treated tooth. It is therefore essential to understand which mechanisms are at the basis of this process to control possible evolutions as underlined by: " Guarnieri R, Miccoli G, Reda R, Mazzoni A, Di Nardo D, Testarelli L. Sulcus fluid volume, IL-6, and Il-1b concentrations in periodontal and peri-implant tissues comparing machined and laser-microtextured collar/abutment surfaces during 12 weeks of healing: A split-mouth RCT. Clin Oral Implants Res. 2022 Jan;33(1):94-104. doi: 10.1111/clr.13868."
Answer 1. We have added the suggested information in the introduction section, as you suggested, in the new version of the manuscript.
2.- Please check figure 4.
Answer 2. Figure 4 was corrected in the new version of the manuscript.
3.- Please check double spaces and typos in the text.
Answer 3. We have incorporated the suggestion of the reviewer, and we correct the typing errors and language editing
Thanks again for your time and consideration of this manuscript, and we look forward to receiving your feedback.
Kind Regards
Alejandra
Round 2
Reviewer 1 Report
Thank you for the corrected version of the manuscript. The text still demands intense language editing. Please take a special attention to the Abstract and Conclusions, like:
“in periodontitis patients, increased expression of miRNA-20a, miRNA-30e, and miRNA-93 but decreased NFAT5 mRNA” – was detected?
„ which should analyze in greater depth in future studies.“ Should be analyzed?
“that encourage analysis of more in-depth miRNAs and hypoxia in the periodontitis
context and to assess the impact of these miRNAs in the pathogenesis and epigenetic modifications that could occur in periodontitis in more extensive studies” ?
and so one.
The expression like “NFAT5 mRNA was significantly lower in periodontitis” is incorrect. Lower was the concentration or the level of mRNA.
It remains unclear which 3 samples were involved in GMSCs analysis and what is presented in Fig 5 D.
I also suggest not to include “3.- GMSCs under hypoxia did not contribute to the modulation of NFAT5 or miRNA-93” in Discussion or correct the expression.
Best regards.
Author Response
Dear Reviewer
Thank you very much again for reviewing our manuscript. I have reviewed and corrected all the suggestions.
Thanks a lot
Kind regards
Alejandra